# A Convenient Approach towards the Synthesis of ADMDP Type Iminosugars and Nojirimycin Derivatives from Sugar-Derived Lactams

**DOI:** 10.3390/molecules26185459

**Published:** 2021-09-08

**Authors:** Piotr Szcześniak, Barbara Grzeszczyk, Bartłomiej Furman

**Affiliations:** Institute of Organic Chemistry, Polish Academy of Sciences, Kasprzaka 44/52, 01-224 Warsaw, Poland; pszczesniak@icho.edu.pl (P.S.); bgrzeszczyk@icho.edu.pl (B.G.)

**Keywords:** sugar-derived lactams, sugar-derived imines, iminosugars, ADMDP, nojirimycin derivatives

## Abstract

An efficient method for the synthesis of nojirimycin- and pyrrolidine-based iminosugar derivatives has been developed. The strategy is based on the partial reduction in sugar-derived lactams by Schwartz’s reagent and tandem stereoselective nucleophilic addition of cyanide or a silyl enol ether dictated by Woerpel’s or diffusion control models, which affords amino-modified iminosugars, such as ADMDP or higher nojirimycin derivatives.

## 1. Introduction

Iminosugars are a large group of carbohydrate analogues that have received a lot of attention due to their ability to inhibit enzymes responsible for the formation or cleavage of glycosidic bonds [1,2,3,4,5]. They have been recognized as potent active agents in the treatment of various diseases, such as diabetes, lysosomal storage disorder, viral infection, and cancer [1,6,7,8,9]. Some drugs based on iminosugars are already in use, while a few others are in clinical trials. For example, Glyset^®^ is used to treat non-insulin-dependent diabetes, and Zavesca^®^ is used in Gaucher’s disease treatment. Another piperidine-originated iminosugar, 1-deoxynojirimycin (DNJ), and its amino derivative (**4b**), displayed similar inhibition to α-glucosidase [10]. α-Homonojirimycin is a powerful α-glucosidase inhibitor and is expected to be a drug candidate for antidiabetic therapy [11].



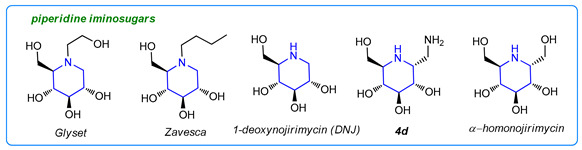



Pyrrolidine-derived iminosugars are also promising drug candidates [1]. For example, 2,5-dihydroxymethyl-3,4-di-hydroxypyrrolidine (DMDP) is a known glycosidase inhibitor. In turn, 1-aminodeoxy-DMDP (ADMDP), an unnatural product that possesses inhibitory activity against *n*-acetyl-β-glucosamidase [12], as well as other DMDP analogues [13,14,15,16], have been found to be potential drug candidates for osteoarthritis [13,14,15,16] and bacterial infections [16,17].



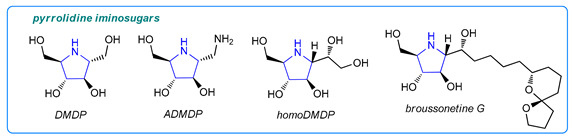



The high potential of amino-iminosugars has resulted in the development of various methods for their synthesis. The first preparation methods of ADMDP and its isomers were concerned with the transformation of naturally occurring DMDP (and its stereo analogues) by selective conversion of one of the side-chain hydroxyl groups into an amino functionality [13].Other strategies include intramolecular cyclization of C2-symmetric amino alcohols [18], transformation of non-racemic γ-lactams [19], nucleophilic ring opening of sugar-derived bis-aziridines [20], and nucleophilic addition of cyanide to sugar-derived cyclic nitrones, followed by reduction [21,22]. Stocker and colleagues [23] reported the synthesis of ADMDP and analogues by carbamate annulation, whereas the group of Ramesh reported their synthesis *via* deamination of glycals [24]. In our previous work, we found that the furanosyl and pyranosyl imines can be obtained from sugar-derived lactams via a Schwartz’s-reagent-mediated partial reduction in amide function [25]. Moreover, we demonstrated that these cyclic imines could be directly used (without isolation) in further transformations [25,26,27], leading to, for example, highly functionalized pyrrolidine and piperidine iminosugars via a tandem one-pot reduction/Grignard reagent or allyltributyltin addition sequence (Figure 1a) [25]. This developed protocol was applied for the preparation of two pyrrolidine derivatives: 6-deoxy-DMDP and radicamine B [25]. Developing our group’s program on reductive sugar-derived lactam functionalization [25,26,27], we report herein the flexible and robust access to amino-iminosugars, such as ADMDP and higher derivatives, via a *gluco*- or *arabino*-lactam reduction/cyanide ion or silyl enol ether addition sequence (Figure 1b).

## 2. Results and Discussion

Our synthesis of amino-modified iminosugars started with the synthesis of initial sugar-derived lactams. We chose *gluco*-**1a** and *arabino*-**1b** lactam, obtained from the corresponding commercially available glucose and arabinose, as model starting materials for further studies (Figure 1) [25,28].

With appropriate substrates (**1a** and **1b**) in hand, we investigated a *one-pot* lactam reduction/cyanide addition reaction (Figure 2). *Arabino*-lactam **1a** was treated with 1.6 equiv of Schwartz’s reagent Cp_2_Zr(H)Cl, leading to appropriate imine **2a** in almost quantitative yield according to the NMR analysis of the crude reaction mixture. The progress of the reaction can easily be followed thanks to simple direct observation: initially white, then heterogeneous mixture turns to a clear solution upon reduction completion (approximately 10 min). Subsequent treatment of the in situ generated imine **2a** with trimethylsilyl cyanide (2.0 equiv) in the presence of TMSOTf (1.0 equiv) gave a mixture of two nitriles **3a** and **2-*epi*-3a** in a ratio of 60:40 and with an overall yield of 88% after two steps (Figure 2). Hydrogenation of **3a** and **2-*epi*-3a** under the conditions reported by Cheng et al. [22] gave **ADMDP** and its 2-epimer (**2-*epi*-ADMDP**) in 73% and 69% yield, respectively (Figure 2). The same approach was successfully applied for the synthesis of the six-membered amino-iminosugar. Thus, treatment of *gluco*-lactam **1b** with Schwartz’s reagent, followed by TMSOTf-mediated addition of trimethylsilyl cyanide gave nitrile **3b** in 65% yield as a single isomer. Its hydrogenolysis under the conditions presented above afforded gluconojirimycin derivative **4b** in 66% yield (Figure 2). Elsewhere, Cheng et al. [22] reported the transformation of nitrile **2-*epi*-3a** to **2-*epi*-DMDP**. We expect that the same strategy can be used for the synthesis of DMDP and α-homonojirimycin starting form **3a** and **3b,** respectively (Figure 2).

Configuration of the newly generated stereochemical center in nitriles 3**a**, 2-epi-3**a**, and 3**b** have been established by NOE experiments. Addition of TMSCN to gluco-imine 2**b** proceeded syn to the BnO substituent at the C-3 position. This is a result of conformational control of the process, according to Woerpel’s model (Figure 2a) [29,30,31,32,33,34]. In the latter case, the nucleophilic addition of TMSCN to *arabino*-imine **2a** occurs with low stereoselectivity, leading to the separable mixture of isomers **3a** and **2-*epi*-3a** in a ratio of 60:40.

The same modest diastereoselectivity was observed by Woerpel in the reaction of TMSCN with cyclic oxocarbenium ions [35]. Woerpel postulated that this effect can be attributed to the high reactivity of the nucleophilic species involved [36]. The silyl cyanides require activation by a nucleophile in solution (such as the counterion of the Lewis acid) to form a pentacoordinate siliconate ion that transfers the cyano group. This activation step is slow relative to nucleophilic addition. A detailed, step-by-step description of this process has been explained by Woerpel in his vital paper from 2006 [35]. It can be assumed that a similar effect applies in the TMSOTf-mediated addition of TMSCN to *arabino*-imine **2a** (Figure 2b).

Subsequently, other types of nucleophiles, such as silyl enol ethers **5a**, **5b**, and **5c**, were examined in the tandem one-pot reduction/nucleophilic addition protocol (Figure 3). *Gluco*-lactam **1b** was selected as a model compound for this part of the study. On the basis of previous examples, a solution of gluco-lactam **1b** was added to a suspension of Schwartz reagent (1.6 equiv) in THF. After the initially white suspension turned clear, the enol silyl ether (2 equiv) and Yb (OTf)_3_ (10 mol%) were added at −25 °C, and the reaction mixture was warmed slowly to room temperature. The desired functionalized piperidines **6a**, **6b**, and **6c** were obtained in 88%, 68%, and 43% yields, respectively, and with a high level of stereoselectivity (Figure 3). The stereochemical structure of piperidines **6a**, **6b**, and **6c** were confirmed by NOE experiments. The stereochemistry between the substituent at C-2 and the benzyloxy group at C-3 was determined to be in a *syn* arrangement for all piperidines **6a**, **6b**, and **6c**. We are dealing here with the same model of stereocontrol as shown in Figure 2a. Obtained compounds **6a**–**c** are attractive building blocks for the *syn* thesis of functionalized nojirimycin derivatives.

## 3. Conclusions

In conclusion, we have presented a convenient route for the synthesis of amino-modified iminosugars, such as ADMDP, based on the reductive activation of sugar-derived lactones and silyl cyanides. The presented method is one of the shortest preparations of ADMDP. In contrast to similar strategies based on nucleophilic addition to cyclic nitrones, our method uses the more easily available and stable lactams. An additional advantage of the current method over nitrone-based strategies [21,22] is the fact that it can also be successfully applied to the synthesis of six-membered analogues of ADMDP (**4d**). The former method is limited due to the more challenging synthesis and limited stability of six-membered nitrones compared to their five-membered counterparts [37]. Moreover, we have presented that other types of nucleophile, such as silyl enol ethers, can be applied, leading to functionalized nojirimycin derivatives.

## 4. Experimental Section

### 4.1. General Information

^1^H NMR and ^13^C NMR spectra were recorded on Varian VNMRS 500 and Varian VNMRS 600 spectrometers, in CDCl_3_, unless otherwise stated, and with TMS used as an internal standard. Chemical shifts (δ) were given in ppm and coupling constants (*J*) were given in Hertz (Hz). Infrared spectra were recorded on an FT-IR Jasco 6200 and FT-IR Spectrum 2000 Perkin Elmer spectrophotometer. High-resolution mass spectra were recorded on an ESI-TOF Mariner Spectrometer, SYNAPT G2-S HDMS, or AMD 604 mass spectrometer. Optical rotations were measured with a Jasco P-2000 polarimeter. Thin-layer chromatography was performed on Merck aluminium sheet Silica Gel 60 F254. Column chromatography was carried out using Merck silica gel (230–400 mesh).

### 4.2. One-Pot Reduction/TMSCN Addition to Sugar-Lactams **1a** and **1b**—General Procedure for The Synthesis of **3a**, **2-Epi-3a**, and **3b**

A solution of sugar lactam **1a** or **1b** (0.5 mmol) in THF (5 mL) was added dropwise to a suspension of Cp_2_Zr(H)Cl (1.6 equiv, 0.8 mmol, 206 mg,) in THF (5 mL) under an argon atmosphere. The mixture was stirred until the white suspension disappeared (ca. 10 min) to form a clear solution. Next, the imine solution was cooled to −25 °C and TMSOTf was added (1.0 equiv, 0.5 mmol, 60 µL). The mixture was stirred for 10 min and TMSCN (2.0 equiv, 1.0 mmol, 125 µL) was added dropwise. The mixture was warmed to room temperature and stirred overnight. The reaction was quenched by addition of aq. NaHCO_3_ and stirred for 30 min. After dilution with Et_2_O (5 mL), the aqueous phase was separated and washed twice with Et_2_O (2 × 5 mL). The combined organic solutions were dried over anhydrous MgSO_4_, filtered, and solvents were removed under reduced pressure. The residue was chromatographed on silica gel to afford the corresponding amine derivatives **3a**, **2-*epi*-3a**, and **3b**.

**(2*R*,3*R*,4*R*,5*R*)-3,4-Bis(benzyloxy)-5-(benzyloxymethyl)pyrrolidine-2-carbonitrile (3a):** (major isomer), colorless oil; 88% (overall, isolated yield for two steps, both isomers); *dr* 60:40 (determined by ^1^H NMR of crude reaction mixture); R*_f_* 0.34 (1:2 AcOEt/hexanes); chromatography (1:3 AcOEt/hexanes); [α]_D_ +23.6 (*c* 0.2 CH_2_Cl_2_); ^1^H NMR (600 MHz, CDCl_3_) δ: 7.38–7.20 (m, 15H), 4.64 (d, *J* 11.9 Hz, 1H), 4.59 (d, *J* 11.9 Hz, 1H), 4.53 (d, *J* 11.9 Hz, 1H), 4.49 (d, *J* 11.9 Hz, 1H), 4.48-4.47 (m, 2H), 4.14-4.09 (m, 2H), 3.93–3.89 (m, 1H), 3.60–3.53 (m, 2H), 3.38–3.33 (m, 1H); ^13^C NMR (151 MHz, CDCl_3_) δ: 137.8, 137.4, 136.9 128.6, 128.5, 128.4, 128.1, 128.0, 127.95, 127.74, 127.72, 117.6, 83.3, 82.8, 73.3, 72.5, 72.0, 70.4, 62.7, 51.2; IR (film) *v*: 3354, 3030, 2924, 2854, 2246, 1495, 1454, 1376, 1364, 1206, 1091, 1072, 733, 695 cm^−1^; HRMS (ESI-TOF) *m*/*z* calculated for C_27_H_29_N_2_O_3_ [M+H^+^] 429.2178, found 429.2184.

**(2*S*,3*R*,4*R*,5*R*)-3,4-Bis(benzyloxy)-5-(benzyloxymethyl)pyrrolidine-2-carbonitrile (2-*epi*-3a)**: (minor isomer), colorless oil; R*_f_* 0.45 (1:2 AcOEt/hexanes); chromatography (1:3 AcOEt/hexanes); [α]_D_–4.7 (*c* 0.85, CH_2_Cl_2_); ^1^H NMR (600 MHz, CDCl_3_) δ: 7.39–7.24 (m, 14H), 4.59 (d, *J* 11.7 Hz, 1H), 4.55 (d, *J* 11.9 Hz, 1H), 4.52 (d, *J* 9.5 Hz, 1H), 4.51–4.45 (m, 3H), 4.3–4.27 (m, 1H), 4.06–4.03 (m, 1H), 3.87–3.83 (m, 1H), 3.62–3.58 (m, 1H), 3.52–3.46 (m, 2H); ^13^C NMR (151 MHz, CDCl_3_) δ: 137.7, 137.5, 136.6, 128.6, 128.4, 128.3, 127.99, 128.0, 127.81, 127.78, 127.7, 119.0, 87.2, 84.0, 73.3, 72.6, 72.2, 69.5, 62.1, 51.7; IR (film) *v*: 3343, 3064, 3031, 2922, 2863, 2240, 1496, 1454, 1363, 1097, 1028, 738, 698 cm^−1^; HRMS (ESI-TOF) *m*/*z* calculated for C_27_H_29_N_2_O_3_ [M+H^+^] 429.2178, found 429.2189.

**(2*S*,3*S*,4*R*,5*R*,6*R*)-3,4,5-tris(benzyloxy)-6-(benzyloxymethyl)piperidine-2-carbonitrile (3b):** white crystals, mp 144°C; 65% (overall, isolated yield for two steps, both isomers); *dr* >95:5 (from 1H NMR of the crude reaction mixture); R*_f_* 0.32 (1:5 AcOEt/hexanes); chromatography (1:7 AcOEt/hexanes); [α]_D_ + 52.6 (*c*, 0.78 CH_2_Cl_2_); ^1^H NMR (600 MHz, CDCl_3_) δ: 7.45–7.26 (m, 17H), 7.23–7.18 (m, 2H), 4.97 (d, *J* 10.7 Hz, 1H), 4.85 (d, *J* 11.1 Hz, 1H), 4.83 (d, *J* 10.8 Hz, 1H), 4.77 (d, *J* 11.9 Hz, 1H), 4.68 (d, *J* 11.9 Hz, 1H), 4.51–4.46 (m, 2H), 4.41 (d, *J* 11.9 Hz, 1H), 4.08 (d, *J* 5.6 Hz, 1H), 3.87–3.81 (m, 1H), 3.69 (dd, *J* 9.4, 2.4 Hz, 1H), 3.59 (dd, *J* 9.4, 5.6 Hz, 1H), 3.36 (dd, *J* 9.2, 7.3 Hz, 1H), 3.27–3.19 (m, 2H); ^13^C NMR (151 MHz, CDCl_3_) δ: 138.4, 138.0, 137.6, 137.5, 128.6, 128.5, 128.41, 128.40, 128.1, 128.0, 127.97, 127.90, 127.87, 127.77, 127.7, 117.5, 84.2, 79.0, 78.6, 76.0, 75.1, 73.3, 73.2, 69.9, 55.6, 49.7; IR (film) *v*: 2909, 2861, 2230, 1453, 1072, 745, 696 cm^−1^; HRMS (ESI-TOF) *m*/*z* calculated for C_35_H_36_N_2_NaO_4_ [M+Na^+^], 571.2573, found 571.2581.

Copies of ^1^H-, ^13^C-NMR, spectra for new compounds: **3a**, **2-*epi*-3a**, and **3b** are available in the Appendix A.

### 4.3. Deprotection/Nitrile Reduction—General Procedure for: **ADMDP**, **2-Epi-ADMDP**, and **4b**

A mixture of **3a**, **2-*epi*-3a**, or **3b** (0.5 mmol), palladium hydroxide (20 mg), and a catalytic amount of acetic acid in methanol (5 mL) was stirred under a hydrogen atmosphere for 48 h. The reaction mixture was filtered through Celite and the filtrate was concentrated. The residue was purified by chromatography on silica gel to afford the corresponding product **ADMDP, 2-*epi*-ADMDP**, and **4b**.

**11-Amino-1,2,5-trideoxy-2,5-imino-D-mannitol (ADMDP):** colorless oil; isolated yield 73%; R*_f_* 0.01 (DCM/MeOH/EtOH/30% aq. NH_3_, 5/2/2/1); chromatography (25% aq. NH_4_OH (37%) in propanol); [α]_D_ +44.1 (*c*, 0.35 H_2_O); agreement with the literature data^12b^; ^1^H NMR (500 MHz, 2% DCl in D_2_O) δ: 4.08 (t, *J* 7.0 Hz, 1H), 4.04 (t, *J* 7.0 Hz, 1H), 3.89 (dd, *J* 12.7, 3.9 Hz, 1H), 3.79 (dd, *J* 12.7, 7.0 Hz, 1H), 3.73 (td, *J* 7.2, 7.0 Hz, 1H), 3.61 (td, *J* 7.0, 3.9 Hz, 1H,), 3.36 (d, *J* 7.2 Hz, 2H); ^13^C NMR (125 MHz, 2% DCl in D_2_O) δ: 76.4, 73.9, 63.4, 58.4, 58.0, 38.6; IR (film) *v*: 3209, 2925, 1603, 1503, 1406, 1123, 1065, 1034, 813 cm^-1^; HRMS (ESI-TOF) *m*/*z* calculated for C_6_H_14_N_2_O_3_ [M+H^+^], 163.1077, found 163.1082.

**2-*****epi*****-ADMDP:** yellow syrup; isolated yield 69%; R*_f_* 0.05 (DCM/MeOH/EtOH/30% aq. NH_3_, 5/2/2/1); chromatography (25% aq. NH_4_OH (37%) in propanol); [α]_D_+12.1 (*c*, 0.65 H_2_O); agreement with the literature data [12b]; ^1^H NMR (600 MHz, D_2_O) δ: 3.09 (dd, *J* 6.3, 13.1 Hz, 1H), 3.13 (q, *J* 5.5 Hz, 1H), 3.24 (dd, *J* 6.0, 13.1 Hz, 1H), 3.59 (q, *J* 6.1 Hz1H), 3.64 (dd, *J* 6.5, 11.6 Hz1H), 3.74 (dd, *J* 4.5, 11.6 Hz, 1H), 3.90 (dd, *J* 3.8, 5.3 Hz, 1H), 4.20 (dd, *J* 3.8, 5.8 Hz, 1H); ^13^C NMR (151 MHz, D_2_O) δ: 77.9, 76.8, 64.0, 62.2, 55.9, 39.2; HRMS calculated for [C_6_H_14_N_2_O_3_ +H^+^] 163.1077, found 163.1079.

**1-Amino-1-deoxy-2,6-dideoxy-2,6-imino-D-glycero-D-ido-heptopyranose (4b)**: thick oil; isolated yield 66%; Rf 0.06 (DCM/MeOH/EtOH/30% aq. NH3, 5/2/2/1); chromatography (25% aq. NH4OH (37%) in propanol); [α]D +3.1 (c, 0.65 H2O); agreement with the literature data [38]; 1H NMR (500 MHz, D2O) δ: 3.99 (dt, J 7.5, 5.3 Hz, 1H), 3.94 (dd, J 9.0, 5.4 Hz, 1H), 3.91 (dd, J 13.0, 4.8 Hz, 1H), 3.86 (dd, J 13.0, 3.3 Hz, 1H), 3.68 (t, J 9.0 Hz, 1H), 3.67 (dd, J 14.0, 7.5 Hz, 1H), 3.58 (dd, J 10.0, 9.0 Hz, 1H), 3.37 (dd, J 14.0 Hz, 5.4 Hz, 1H), 3.26 (ddd, J 10.0, 4.8, 3.3 Hz, 1H); 13C NMR (125 MHz, D2O) δ: 72.8, 69.3, 68.2, 57.9, 57.0, 52.9, 36.7; IR (film) v: 3384, 1618, 1100, 1033, 900, 838 cm-1; HRMS calculated for [C7H16N2O4+H+] 193.1188, found 193.1182. 

### 4.4. One-Pot Reduction/Enol Silyl Ether Addition to Sugar-Lactams **1b**—General Procedure for the Synthesis of **6a**, **6b**, and **6c**

A solution of sugar lactam **1b** (0.5 mmol) in THF (5 mL) was added dropwise to a suspension of Cp_2_Zr(H)Cl (1.6 equiv, 0.8 mmol, 206 mg) in THF (5 mL) under an argon atmosphere. The mixture was stirred until the white suspension disappeared (ca. 10 min) and a clear solution was formed. Next, the imine solution was cooled to −25 °C and Yb (OTf)_3_ (10 mol%, 0.05 mmol, 31 mg) was added. The mixture was stirred for 10 min and a solution of enol silyl ether (2.0 equiv 1.0 mmol) in THF (5 mL) was added dropwise. The mixture was gradually warmed to ambient temperature and stirred overnight. Then, the reaction was quenched by the addition of aq. NaHCO_3_ and stirred for 30 min. After dilution with Et_2_O (5 mL), the aqueous phase was separated and washed twice with Et_2_O (2 x 5mL). The combined organic layers were dried over anhydrous MgSO_4_, filtered, and the solvents were removed under reduced pressure to afford the residue, which was chromatographed on silica gel to afford the corresponding product **6a**, **6b**, or **6c**.

**(*E*)-ethyl-4-((2R,3S,4R,5R,6R)-3,4,5-tris(benzyloxy)-6-(benzyloxymethyl)piperidin-2-yl)but-2-enoate (6a):** colorless oil; 88% (overall, isolated yield for two steps, both isomers); *dr* >95:5 (determined by ^1^H NMR of crude reaction mixture); colorless oil; R*_f_* 0.22 (1:3 AcOEt/hexanes); chromatography (1:3 AcOEt/hexanes); ^1^H NMR (600 MHz, CDCl_3_) δ: 7.36–7.26 (m, 18H), 7.20–7.18 (m, 2H), 6.95 (ddd, *J* 15.3, 8.5, 6.4 Hz, 1H), 5.90 (d, *J* 15.6 Hz, 1H), 4.94 (d, *J* 10.9 Hz, 1H), 4.85 (d, *J* 10.8 Hz, 1H), 4.80 (d, *J* 10.9 Hz, 1H), 4.69 (d, *J* 11.6 Hz, 1H), 4.64 (d, *J* 11.6 Hz, 1H), 4.50 (d, *J* 9.2 Hz, 1H), 4.48 (d, *J* 8.1 Hz, 1H), 4.41 (d, *J* 11.9 Hz, 1H), 4.22–4.13 (m, 2H), 3.73–3.68 (m, 2H), 3.65–3.62 (m, 1H), 3.54 (dd, *J* 9.0, 5.8 Hz, 1H), 3.43–3.38 (m, 1H), 3.34–3.28 (m, 1H), 2.98–2.91 (m, 1H), 2.66–2.59 (m, 1H), 2.52–2.44 (m, 1H), 1.27 (t, *J* 7.1 Hz, 3H); ^13^C NMR (151 MHz, CDCl_3_) δ: 166.3, 146.5, 138.8, 138.3, 138.0, 128.43, 128.40, 128.36, 128.02, 127.92, 127.81, 127.76, 127.73, 127.70, 127.63, 127.56, 123.6, 83.1, 81.8, 80.2, 75.6, 75.2, 73.2, 72.8, 69.9, 60.2, 53.4, 52.8, 29.7, 28.8, 14.2; IR (film) *v*: 2922, 2860, 1717, 1454, 1096, 1068, 736, 697 cm^−1^; HRMS (ESI-TOF) m/z calculated for C_40_H_48_NO_6_ [M+H^+^] 636.3325, found 636.3329.

**(S)-5-((2R,3S,4S,5R,6R)-3,4,5-tris(benzyloxy)-6-(benzyloxymethyl)piperidin-2-yl)furan-2(5H)-one (6b):** (major isomer), colorless oil; 68% (overall, isolated yield for two steps, both isomers); *dr* 75:25 (determined by ^1^H NMR of crude reaction mixture); colorless oil; R*_f_* 0.43 (1:1 AcOEt/hexanes); chromatography (1:3 AcOEt/hexanes than 1:1 AcOEt/hexanes); ^1^H NMR (600 MHz, CDCl_3_) δ: 7.68–7.66 (m, 1H), 7.36–7.24 (m, 18H), 7.20–7.17 (m, 2H), 6.06 (dd, *J* 5.7, 1.8 Hz, 1H), 5.48 (d, *J* 7.7 Hz, 1H), 4.94 (d, *J* 10.9 Hz, 1H), 4.86 (d, *J* 7.3 Hz, 1H), 4.84 (d, *J* 7.4 Hz, 1H), 4.76 (d, *J* 11.5 Hz, 1H), 4.60 (d, *J* 11.5 Hz, 1H), 4.53–4.49 (m, 2H), 4.43 (d, *J* 12.0 Hz, 1H), 3.87 (d, *J* 8.7 Hz, 1H), 3.84 (d, *J* 5.6 Hz, 1H), 3.60 (dd, *J* 9.5, 2.5 Hz, 1H), 3.48 (dd, *J* 9.5, 6.0 Hz, 1H), 3.43 (t, *J* 9.1 Hz, 1H), 3.23–3.20 (m, 1H), 3.19–3.15 (m, 1H); ^13^C NMR (151 MHz, CDCl_3_) δ: 172.6, 156.6, 138.4, 138.1, 137.8, 137.6, 128.5, 128.42, 128.40, 128.3, 128.0, 127.95, 127.94, 127.88, 127.86, 127.72, 127.70, 127.68, 120.8, 83.3, 81.2, 80.2, 79.8, 75.6, 75.0, 73.8, 73.1, 69.8, 58.2, 54.3; IR (film) *v*: 3030, 2865, 1757, 1454, 1091, 738, 698 cm^−1^; HRMS (ESI-TOF) m/z calculated for C_38_H_40_NO_6_ [M + H^+^] 602.2856, found 606.2856; (**2-*epi*-6b**) (minor isomer) selected signals: ^1^H NMR (600 MHz, CDCl_3_) δ: 7.70 (d, *J* 5.4 Hz, 1H), 6.10 (dd, *J* 5.7, 1.9 Hz, 1H), 5.30 (d, *J* 6.5 Hz, 1H); ^13^C NMR (151 MHz, CDCl_3_) δ: 173.0, 156.2, 138.1, 138.0, 137.9, 137.7, 121.4, 55.2, 54.8.

**1-(4-Fuorophenyl)-2-((2R,3S,4R,5R,6R)-3,4,5-tris(benzyloxy)-6-(benzyloxymethyl)piperidin-2-yl)ethanone (6c):** colorless oil; 43% (overall, isolated yield for two steps, both isomers); *dr* > 95:5 (determined by ^1^H NMR of crude reaction mixture); R*_f_* 0.32 (1:3 AcOEt/hexanes); chromatography (1:4 AcOEt/hexanes); ^1^H NMR (600 MHz, CDCl_3_) δ: 8.03 7.89 (m, 2H), 7.4–7.23 (m, 18H), 7.20–7.16 (m, 2H), 7.11–7.05 (m, 2H), 4.96 (d, *J* 10.8 Hz, 1H), 4.85 (d, *J* 10.5 Hz, 1H), 4.82 (d, *J* 10.9 Hz, 1H), 4.66 (d, *J* 11.5 Hz, 1H), 4.63 (d, *J* 3.8 Hz, 1H), 4.50–4.41 (m, 3H), 4.10–4.05 (m, 1H), 3.80–3.76 (m, 1H), 3.74–3.69 (m, 1H), 3.61 (dd, *J* 9.1, 2.3 Hz, 1H), 3.53–3.49 (m, 1H), 3.45–3.39 (m, 1H), 3.33 (dd, *J* 16.9, 3.5 Hz, 1H), 3.18 (dd, *J* 16.9, 9.5 Hz, 1H), 3.00–2.94 (m, 1H); ^13^C NMR (151 MHz, CDCl_3_) δ: 197.6, 165.75 (d, *J* = 254.8 Hz), 138.7, 138.2, 138.1, 137.9, 130.75 (d, *J* = 9.3 Hz), 128.41, 128.37, 128.34, 128.0, 127.9, 127.84, 127.77, 127.75, 127.71, 127.65, 127.57, 115.64 (d, *J* = 21.9 Hz), 75.6, 75.3, 73.2, 72.9, 53.5, 50.6, 34.5; IR (film) *v*: 3031, 2918, 2862, 1681, 1597, 1453, 1230, 1095, 1068, 737, 698 cm^−1^; HRMS (ESI-TOF) *m*/*z* calculated for C_42_H_43_NO_5_F [M+H^+^] 660.3125, found 660.3124.

Copies of ^1^H-, ^13^C-NMR, spectra for new compounds: **6a, 6b,** and **6c** are available in the Appendix A.

## Data Availability

The data is not available from the authors.

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
