# Peer review of "A Convenient Approach towards the Synthesis of ADMDP Type Iminosugars and Nojirimycin Derivatives from Sugar-Derived Lactams"

_molecules, 2021, doi:10.3390/molecules26185459_

Round 1

Reviewer 1 Report

The authors proposed a convenient way to synthesize amino-modified imino sugars (such as ADMDP) based on the reductive activation of sugar-derived lactones and silyl cyanide. The proposed method is one of the shortest preparation methods for ADMDP. This work has obtained detailed data, which provides a reference to future researchers in this field. However, some severe issues need to be addressed before considering acceptance.

However, I cannot recommend to accept this article in current version unless the authors address the following concerns.

  1. In the second section, the reactants equivalent and yield are given, but the scale of the reactions are not mentioned. Are they milligrams, grams, or kilograms?
  2. In the Experimental Section, the authors present chromatographic characterization of various reaction products. We can consider presenting these data in images instead of directly listing them, which are not intuitive.
  3. Some relative papers may enrich the concept and background of molecular level engineering as references: Nano Energy, 2021, 89, 106321; Adv. Mater., 2021, 2101262. (doi.org/10.1002/adma.202101262); Adv. Funct. Mater., 2021, 31, 2010962; Tetrahedron, 2021, 79: 131837; Pharmaceuticals 2019, 12(3), 108; Org. Lett. 2017, 19, 16, 4403–4406

Author Response

hgyfgg

Reviewer 2 Report

The present manuscript entitled “A convenient approach towards the synthesis of ADMDP type iminosugars and nojirimycin derivatives from sugar-derived lactams” by Furman and coworkers describes a method for the nucleophilic addition to the gluco- and arabino lactams. Compared  to the reported methodologies, the present methodology is extended to six-membered analogue as well. The outline of the work is clearly presented in the manuscript, and the manuscript is well-written. Thus, I would recommend the publication of this manuscript in Molecules after addressing the following comments.

Comments:

  1. The number of authors mentioned in the manuscript and in the Supporting Information file are different. Please correct this issue .
  2. Although the characterization data for ADMDP, 2-epi-ADMDP and 4b are provided in the manuscript, I do not find their NMR spectra in the Supporting Information file. Please provide the NMR spectra of these compounds for the benefit of the readers.
  3. Please provide a caption for Scheme 1.
  4. I find some inconsistencies in the characterization data and the NMR spectra. For example, the 1H NMR spectrum of 2-epi-3a shows 14 protons within 7.36-7.25 ppm range, but it is differently mentioned in its characterization data. Similar inconsistency can be found for compound 3b as well. Please correct these issues.

Minor typos:

  1. The citation number for reference 1 should be superscript.
  2. Page 3, line 76, please change “Cheng et all” into “Cheng et al.”
  3. Page 4, line 116, compound number 1b should be in bold.

Author Response

 Referee: 2
Comments to the Author
1) The number of authors mentioned in the manuscript and in the Supporting Information file are different. Please correct this issue .
The number of authors and order has been corrected.
2) Although the characterization data for ADMDP, 2-epi-ADMDP and 4b are provided in the
manuscript, I do not find their NMR spectra in the Supporting Information file. Please provide the NMR spectra of these compounds for the benefit of the readers.
We did not include the spectra of compounds ADMDP, 2-epi-ADMDP and 4b because they are known, we have given references to original works (ref.12b and 22)
3) Please provide a caption for Scheme 1.
The caption for scheme 1 has been provided.
4) I find some inconsistencies in the characterization data and the NMR spectra. For example, the 1H NMR spectrum of 2-epi-3a shows 14 protons within 7.36-7.25 ppm range, but it is differently mentioned in its characterization data. Similar inconsistency can be found for compound 3b as well. Please correct these issues.

We corrected the number of protons [7.39–7.24 (m, 14H) for 2-epi-3a] and [7.45–7.26 (m, 17H for 3b) in the description of collected spectral data. The difference between the theoretical number of protons and that observed in the spectrum is due to overlapping of aromatic signals.
5) Minor typos:
- The citation number for reference 1 should be superscript.
- Page 3, line 76, please change “Cheng et all” into “Cheng et al.”
- Page 4, line 116, compound number 1b should be in bold.
All typo errors have been corrected and marked in the text.

Reviewer 3 Report

I was set to write a favorable review, but first wanted to check regarding whether there was evidence for diffusion control.  I didn’t see any in the manuscript being evaluated, so I checked Woerpel’s studies and found the following on page 8687 of JACS, 2006, 128, 8671-8677 (cited as ref 19 in the manuscript under review):

Once formed, however, the siliconate ion reacts with the electrophile at rates at or near the rate of diffusion. When reactions approach the diffusion rate limit, stereoelectronic effects cannot control the reaction outcome, because the stereochemistry-determining step occurs before bond formation. Stereoelectronic control over these reactions can be re-established by either stabilizing the charged intermediates (both the nucleophile and electrophile) in polar media or by structurally attenuating the electrophilicity of the oxocarbenium ion intermediate (eqs 4 and 5).

This sounded familiar, so I returned to the manuscript under review and found on the 3rd page a direct COPY from the Woerpel manuscript.  Please compare the following – from the manuscript under review – with that highlighted above from Woerpel’s paper. 

Once formed, however, the siliconate ion reacts with the electrophile at rates at or near the rate of diffusion. When reactions approach the diffusion rate limit, stereoelectronic effects cannot control the reaction outcome, because the stereochemistry-determining step occurs before bond formation. Stereoelectronic control over these reactions can be re-established by either stabilizing the charged intermediates (both the nucleophile and electrophile) in polar media or by structurally attenuating the electrophilicity of the oxocarbenium ion intermediate.

Given this, I suggest that the manuscript be returned to the authors. These serious issues must be addressed before any additional review can take place.

Author Response

 Referee: 3
Comments to the Author
I was set to write a favorable review, but first wanted to check regarding whether there was evidence
for diffusion control. I didn’t see any in the manuscript being evaluated, so I checked Woerpel’s
studies and found the following on page 8687 of JACS, 2006, 128, 8671-8677 (cited as ref 19 in the
manuscript under review):
Once formed, however, the siliconate ion reacts with the electrophile at rates at or near the rate of
diffusion. When reactions approach the diffusion rate limit, stereoelectronic effects cannot control the
reaction outcome, because the stereochemistry-determining step occurs before bond formation.
Stereoelectronic control over these reactions can be re-established by either stabilizing the charged
intermediates (both the nucleophile and electrophile) in polar media or by structurally attenuating
the electrophilicity of the oxocarbenium ion intermediate (eqs 4 and 5).
This sounded familiar, so I returned to the manuscript under review and found on the 3rd page a
direct COPY from the Woerpel manuscript. Please compare the following – from the manuscript
under review – with that highlighted above from Woerpel’s paper.
Once formed, however, the siliconate ion reacts with the electrophile at rates at or near the rate of
diffusion. When reactions approach the diffusion rate limit, stereoelectronic effects cannot control the
reaction outcome, because the stereochemistry-determining step occurs before bond formation.
Stereoelectronic control over these reactions can be re-established by either stabilizing the charged
intermediates (both the nucleophile and electrophile) in polar media or by structurally attenuating
the electrophilicity of the oxocarbenium ion intermediate.
Given this, I suggest that the manuscript be returned to the authors. These serious issues must be
addressed before any additional review can take place.

We thank the Referee for this remark.
We apologize for our oversight, of course a copy of the
Woerpel’s studies should not be included in the text. On the basis of our research, we found that the Woerpel’s model can be used to explain the stereochemistry of the nucleophilic addition to sugar derivative cyclic imines. When preparing the manuscript, we collected materials that we wanted to include in the paper (placing texts from
original works as temporary placeholders for intended narrative). The manuscript was read and revised by many people, and finally we overlooked that this text was taken from the original work. The original
Woerpel’s text has been removed from the publication. In this place we added information ‘’A detailed, step-by-step description of this process has been explained by Woerpel, in
his vital paper from 2006.
19’’

Round 2

Reviewer 2 Report

The authors of this manuscript “A convenient approach towards the synthesis of ADMDP type iminosugars and nojirimycin derivatives from sugar-derived lactams” have successfully addressed all the issues raised by this reviewer. The authors have made necessary improvements to meet the requirements of publication. Thus, this reviewer approves this revised manuscript for publication in Molecules.

Reviewer 3 Report

The authors have satisfactorily addressed my concern.  The manuscript is now suitable for publication.